# Multimodal analysis of RNA sequencing data powers discovery of complex trait genetics

Daniel Munro [1,2,3], Nava Ehsan [3], Seyed Mehdi Esmaeili-Fard [2], Alexander Gusev [4,7] ✉, Abraham A. Palmer [1,5,7] ✉ & Pejman Mohammadi [2,3,6,7] ✉

RNA sequencing has the potential to reveal many modalities of transcriptional regulation, such as various splicing phenotypes, but studies on gene regulation are often limited to gene expression due to the complexity of extracting and analyzing multiple RNA phenotypes. Here, we present Pantry, a framework to efficiently generate diverse RNA phenotypes from RNA sequencing data and perform downstream integrative analyses with genetic data. Pantry generates phenotypes from six modalities of transcriptional regulation (gene expression, isoform ratios, splice junction usage, alternative TSS/polyA usage, and RNA stability) and integrates them with genetic data via QTL mapping, TWAS, and colocalization testing. We apply Pantry to Geuvadis and GTEx data, finding that 4768 of the genes with no identified eQTL in Geuvadis have QTL in at least one other transcriptional modality, resulting in a 66% increase in genes over eQTL mapping. We further found that the QTL exhibit modality-specific functional properties that are further reinforced by joint analysis of different RNA modalities. We also show that generalizing TWAS to multiple RNA modalities approximately doubles the discovery of unique gene-trait associations, and enhances identification of regulatory mechanisms underlying GWAS signal in 42% of previously associated gene-trait pairs.

RNA sequencing is used to quantify transcriptomic activity, and can be combined with genotyping to detect heritable differences in gene regulation. This quantification often includes only total gene expression and, less often, some form of alternative splicing phenotype, such as intron excision rates, resulting in expression quantitative trait loci (eQTLs) and splice QTLs (sQTLs) or predicted expression models. These molecular phenotypes can provide evidence for mechanisms by which heritable differences in gene regulation serve as the molecular intermediates between GWAS association signals and complex traits[1,2]. Other forms of transcriptomic variation, such as alternative transcription start site (TSS), alternative polyadenylation (polyA), and splice isoform ratios, have been found to explain an additional portion[3,4]. Importantly, all of these phenotypes are based on RNA-seq data but require multiple different analytic methods.

Methods and resources already exist to identify genetically driven associations between these RNA phenotypes and complex traits[5-9]. However, it is common that only gene expression is examined because of the significant extra effort needed to obtain other RNA phenotypes. Difficulties include data formatting issues, software dependencies, post-processing, computational resources, lack of field expertise, and other practical considerations. Another challenge is the statistical complexity of interpreting these correlated RNA phenotypes and

[1]Department of Psychiatry, UC San Diego, La Jolla, CA, USA. [2]Center for Immunity and Immunotherapies, Seattle Children's Research Institute, Seattle, WA, USA. [3]Department of Integrative Structural and Computational Biology, Scripps Research, La Jolla, CA, USA. [4]Division of Population Sciences, Dana-Farber Cancer Institute and Harvard Medical School, Boston, MA, USA. [5]Institute for Genomic Medicine, UC San Diego, La Jolla, CA, USA. [6]Department of Pediatrics, University of Washington School of Medicine, Seattle, WA, USA. [7]These authors jointly supervised this work: Alexander Gusev, Abraham A. Palmer, Pejman Mohammadi. ✉e-mail: alexander_gusev@dfci.harvard.edu; aap@ucsd.edu; pejmanm@uw.edu

downstream results in aggregate. For example, it is not straightforward to distinguish a case where two genetic association signals in different RNA modalities reveal two biological mechanisms from a case where a single mechanism is reflected in two related RNA modalities.

We present Pantry, a framework for pan-transcriptomic phenotyping that streamlines the quantification of multiple RNA modalities and their use in downstream applications, including molecular QTL (xQTL) mapping and transcriptome-wide association studies (TWAS). We apply all of this to 50 human tissue datasets and demonstrate that when TWAS is generalized to include multiple RNA modalities (xTWAS) there is a substantial increase in the number of significant gene-trait associations, and improved specification of the most relevant RNA phenotype.

## Results

We developed Pantry, an end-to-end framework for multimodal analysis of RNA-seq data from populations for genomic interpretation (Fig. 1a). Currently, Pantry encompasses six modalities of regulatory variation. Two of these, total gene expression and RNA stability, result in one phenotype per gene, while the other four can produce multiple molecular phenotypes per gene, such as relative abundance of each unique transcript isoform. We generated data on these six modalities of transcriptome regulation for 445 lymphoblastoid cell line (LCL)

samples in Geuvadis[10] and all 17,350 samples across 54 tissues in the GTEx Project V8 release[1]. We limited our analysis to protein-coding genes and lncRNAs. We generated 204,273 phenotypes per sample, spanning 25,657 genes in Geuvadis data (Table 1, Fig. 1b, c), and similar figures in individual GTEx tissues (Supplementary Fig. 1).

### Applying RNA phenotypes to genetic analyses

**Mapping xQTLs across multiple modalities increases xGene discovery.** To identify genetic determinants of individual transcriptome phenotypes generated by Pantry, we developed Pheast (PHEnotype Application STreamlined). Pheast uses an approach previously used for splice QTL mapping to simultaneously map cis-QTLs across all six transcriptome modalities; we refer to this as cross-modality mapping. Specifically, a stepwise regression procedure implemented in tensorQTL that is used to identify conditionally independent cis-QTLs can be applied to grouped phenotypes, such as multiple splice phenotypes per gene[11,12]. But phenotypes of different modalities could also be correlated and produce redundant xQTLs, such as when alternative splicing (measured as intron excision ratio) alters the isoform ratios or total gene expression estimates. While some xQTL overlap between modalities could reflect mechanistically distinct effects, it can be useful to enumerate independent genetic signals. Pantry Pheast combines the sets of phenotypes across all modalities and maps cis-QTLs

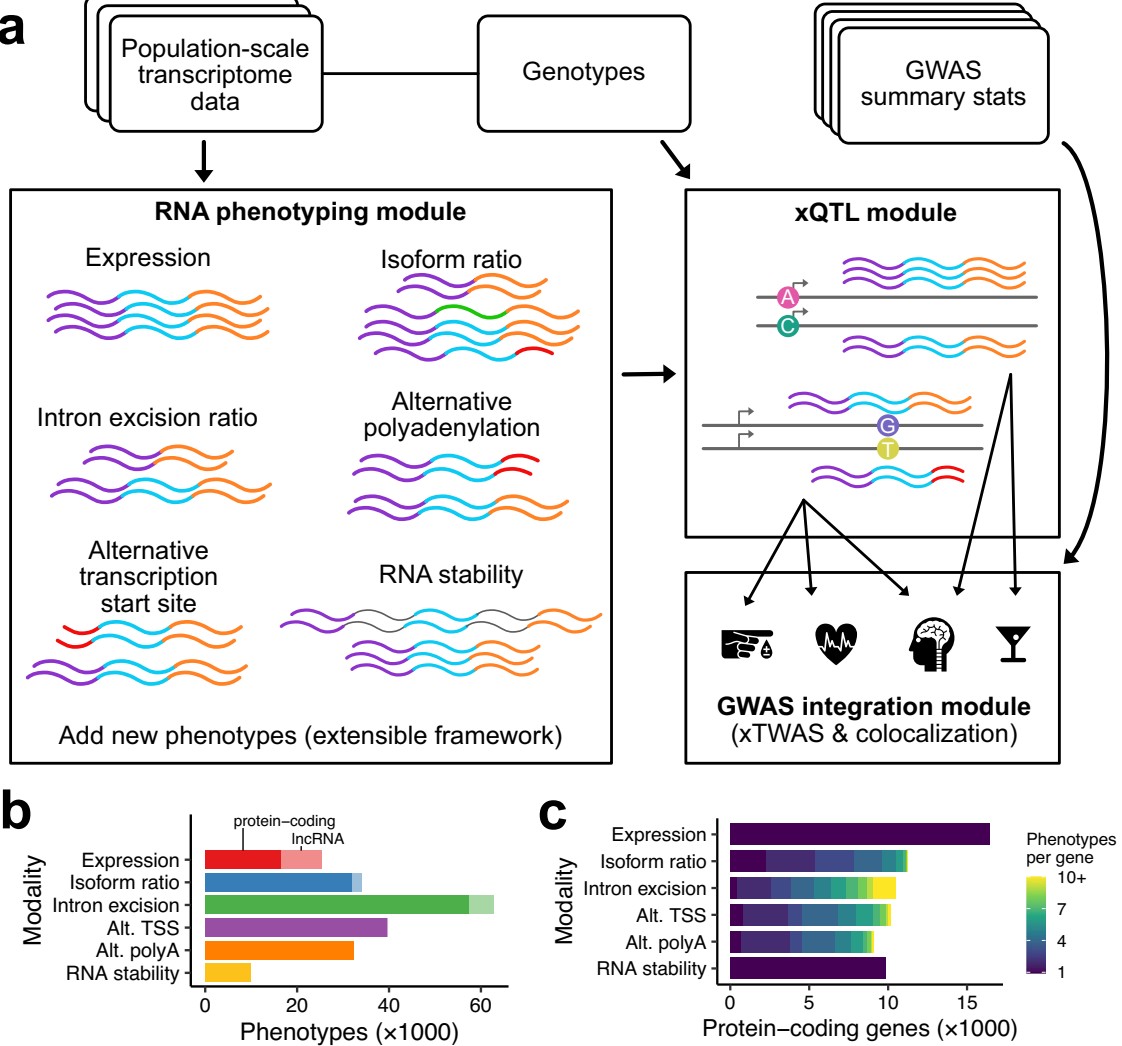

**Fig. 1 | Pantry's multimodal RNA phenotyping. a** Flow chart of the Pantry procedure. **b** Number of phenotypes extracted per modality for the Geuvadis dataset. Bars are colored by modality and the shading indicates the gene type: protein-coding gene or lncRNA. **c** Number of protein-coding genes represented by the phenotypes in (**b**), and the number of phenotypes extracted per gene in each modality. Source data are provided as a Source Data file.

**Table 1 | Default RNA modalities included in Pantry and statistics for its application to the Geuvadis dataset**

| Modality | Method | No. phenotypes produced | No. genes with phenotypes | No. phenotypes per gene (mean ± S.D.) |
|---|---|---|---|---|
| Total gene expression | kallisto[40] | 25,311 | 25,311 | 1±0 |
| Isoform ratio | kallisto, per-isoform count over sum of isoform counts per gene | 34,163 | 12,344 | 2.8±1.5 |
| Intron excision ratio | Count intron junctions using regtools[41], cluster using leafcutter[26] | 62,870 | 11,901 | 5.3±3.8 |
| Alternative TSS | Generate alternative TSS annotations with txrevise[3], quantify with kallisto | 39,632 | 10,214 | 3.9±2.1 |
| Alternative polyA | Generate alternative polyA annotations with txrevise, quantify with kallisto | 32,411 | 9,100 | 3.6±1.9 |
| RNA stability | featureCounts[42], constitutive exon read count to intron read count ratio per gene[43] | 9886 | 9886 | 1±0 |
| Total | N/A | 204,273 | 25,657 | 8.0±8.3 |

Genes include protein-coding genes and lncRNAs. Source data are provided as a Source Data file.

with stepwise regression, considering all phenotypes per gene as a single group. This is implemented using tensorQTL's stepwise regression, grouped phenotype, and data permutation features (Supplementary Methods). All xQTL results hereafter refer to those from this cross-modality cis-xQTL mapping strategy unless otherwise noted.

Using the 445 Geuvadis samples, we identified 21,045 conditionally independent xQTLs for 11,983 genes across the six studied modalities. Expression QTLs were the most abundant, with eQTLs found for 7215 genes, which was more than 3.2 times greater than isoform ratio, which was the second-most abundant xQTL group (Fig. 2a). However, for 4768 genes with no identified eQTL, we found xQTLs in at least one of the other modalities. This represents a 66% increase in the number of xQTL genes (xGenes), highlighting the utility of analyzing multiple modalities of transcriptional regulation. Multiple conditionally independent xQTLs were found for 42.7% of xGenes (Fig. 2b). The xQTLs for each gene were ranked in order of detection by stepwise regression. The proportion from each modality varied across ranks such that stronger xQTLs were most likely to be for expression, and subsequent xQTLs were more likely to be for isoform ratio or intron excision ratio (Fig. 2c). This trend could be influenced by the relative strength of the true genetic signals in each modality, the power to detect the signals with each method, and differences in the number of phenotypes per gene.

We similarly mapped xQTLs for each of 49 GTEx tissues, separately per modality (Supplementary Data 1) and with cross-modality mapping (Supplementary Data 2). We discovered comparable numbers of xQTLs as for Geuvadis, which varied across tissues due to factors such as sample size, but generally found non-expression xQTLs in thousands of genes per tissue for which no eQTLs were found in our data, resulting in a 71% increase of xGenes over eGenes alone on average (Supplementary Fig. 2).

To measure concordance of xQTLs between independent datasets, for each modality we identified the strongest xQTL per xGene in Geuvadis, and extracted the associations for the same variant-RNA phenotype pairs, if tested, in GTEx EBV-transformed lymphocytes (LCL). This resulted in 21,345, or 78%, of the Geuvadis pairs that could be compared between the datasets. We found that the regression slopes were consistent in both direction and magnitude between the two datasets, with Pearson correlation coefficients ranging from 0.80 to 0.89 per modality and mean Deming regression slope of 1.005 (Supplementary Fig. 3).

**Location and functional effect of xVariants reflect their associated modality.** While we used the same cis- window of ±1 Mb from the transcription start site to map xQTLs for all six modalities, we found that the location of the mapped xQTLs relative to their xGene varies strongly depending on the modality (Fig. 2d, Supplementary Fig. 4). As expected, the distributions of expression and alternative TSS xQTLs peak around the start site, while the distribution of alternative polyA xQTLs peak around the end. Isoform ratio, intron excision ratio, and RNA stability xQTLs are more uniformly distributed across the length of their genes.

We examined functional annotations for each xQTL top variant (xVariant) to identify which annotations were most enriched in each RNA modality. Results were largely in line with expectations. For example, splicing annotations were most enriched in the intron excision ratio xVariants, 5′ UTR variants and promoters were most enriched in alternative TSS xVariants, and 3′ UTR variants were most enriched in alternative polyA xVariants (Fig. 2e). Expression was the second-most enriched modality in 5′ UTR and promoter variants, and the most enriched modality for promoter-flanking variants. These enrichment levels were largely consistent across GTEx tissues.

**Cross-modality mapping reduces redundancy of xQTLs.** We analyzed the impact of Pantry's cross-modality xQTL mapping strategy compared to the conventional method of mapping conditionally

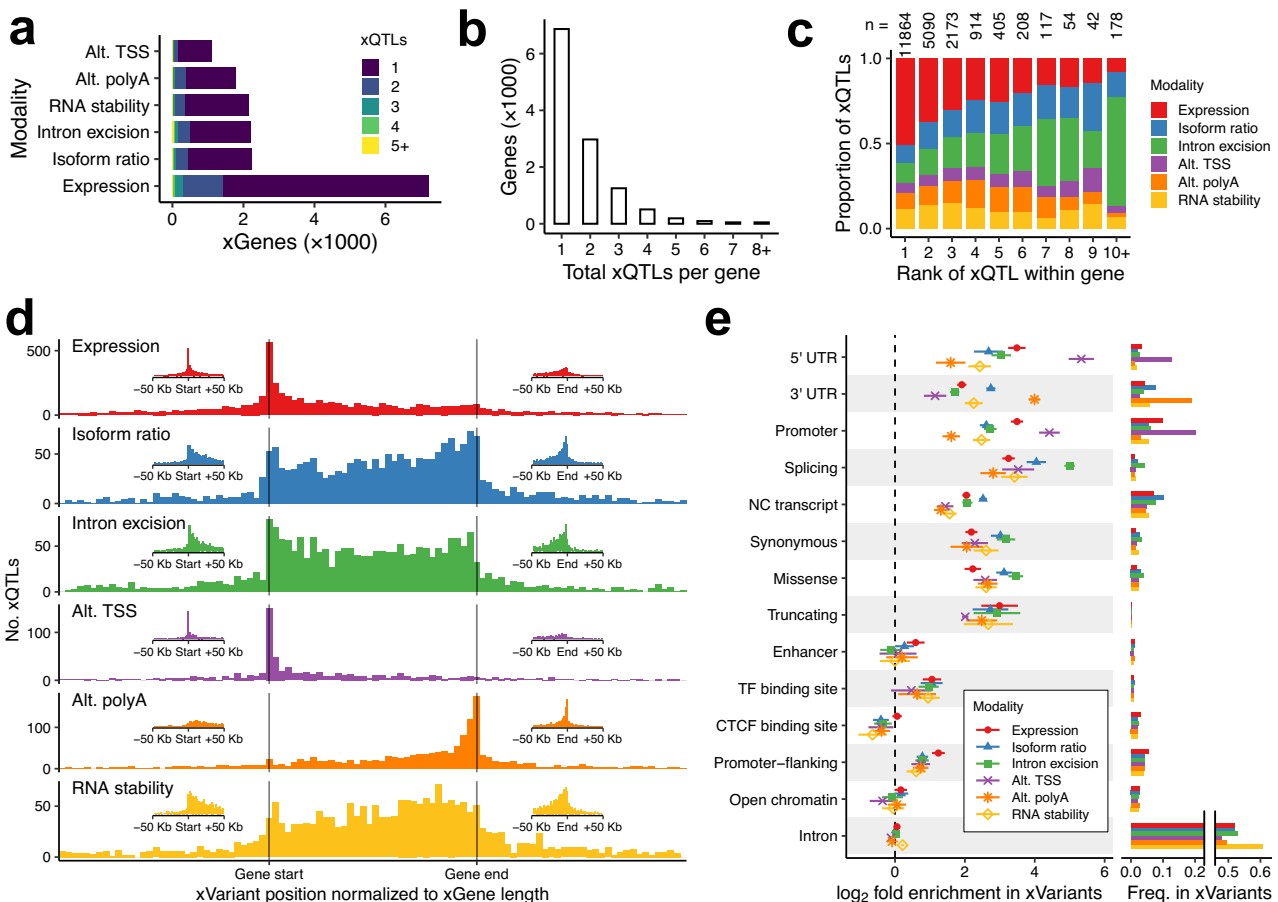

**Fig. 2 | Multimodal xQTL mapping. a** For each modality, the number of xGenes found in Geuvadis, colored according to the number of xQTLs found for the gene. These xQTLs were produced from a single cross-modality mapping, and are grouped by modality here for visual comparison. **b** Total xQTLs per xGene in Geuvadis, summed over modalities. **c** For each xGene, Geuvadis xQTLs were ranked by association strength, and the proportion of xQTLs per rank that belong to each modality is shown. The number of xQTLs in each column are shown above the columns. **d** Location of xQTLs in Geuvadis, relative to their xGene. The genomic coordinates of each xGene and that gene's xQTLs were linearly transformed such that the gene starts and ends are aligned on the x-axis. Only the 65% of xQTLs within one gene's length of the xGene start or end are shown. The insets show the distribution of xQTL positions within 50 Kb of the xGene start or end, without normalizing by gene length. The two insets per modality have the same y-axis scale. Histograms are colored by modality. **e** Enrichment of functional annotations in the top variants of each xQTL for each RNA modality, relative to all variants tested for xQTLs for each modality, for all 49 GTEx tissues. Points and horizontal segments show mean and standard deviation across tissues, respectively. Annotation categories are ordered by decreasing variance of their six $\log_2$ enrichment means. Bar plots show, for each annotation, the proportion of xVariants in each modality with the annotation, averaged across tissues. Variants can be assigned more than one annotation. Source data are provided as a Source Data file.

independent QTLs separately per modality. Cross-modality mapping resulted in fewer total xQTLs per gene on average (1.76 in Geuvadis) compared to 2.94 for separate-modality mapping (Fig. 3a). This general trend is expected and desirable because the goal of cross-modality mapping is to eliminate correlated signals. Notably, we only observe a slight decrease (10.4%) in the total number of xGenes in spite of the 46.4% decrease in the total number of xQTLs (Fig. 3b). Looking at individual modalities, however, we see a drastic drop (median 39.2%) in the number of xGenes (Fig. 3b). This pattern points to deconvolution of confounding xQTL effects observed in multiple modalities by the cross-modality mapping strategy. To this end, we looked specifically at the consistency of expression QTL effect sizes. Using data from GTEx subcutaneous adipose tissue, we measured allelic fold change (aFC) from gene expression data and again from allele-specific expression (ASE) data. These two measurements of cis-regulatory effect size are largely affected by independent sources of noise and as such allow us to gauge the quality of mapped cis-eQTLs[13]. The Pearson correlation between the two aFC measures was slightly higher for cross-modality mapping ($r = 0.721$, 95% CI [0.709, 0.733]) than for p-value-matched eQTLs from separate-modality mapping ($r = 0.703$, 95% CI [0.690,

0.715]), suggesting a refinement of eQTL signals (Supplementary Fig. 5). We note, however, that in cases where a variant causes mechanistically distinct effects on multiple modalities, this cross-modality mapping procedure may also omit the xQTLs for some of those modalities.

Next, we looked at how cross-modality mapping affects the overall functional characteristics of the resulting set of xVariants. For genes with Geuvadis xQTLs from both mapping methods, we examined, for each modality, the subset of genes that had significant xQTLs using separate-modality mapping but no significant xQTLs when using cross-modality mapping ("removed"; Fig. 3b) and the subset that were also found with cross-modality mapping ("retained"; Fig. 3b). We hypothesize that the removed xQTLs were found in multiple modalities and were better characterized with a phenotype of a different modality. We observed sharper modality-specific distributions of xVariant positions in the retained xQTLs (Fig. 3c). Specifically, there were relatively fewer retained expression xVariants within the gene body and especially near the transcription end site (TES), compared to the peak at the TSS, and likewise fewer alternative TSS and polyA xVariants within the gene body relative to the peak at the TSS or TES,

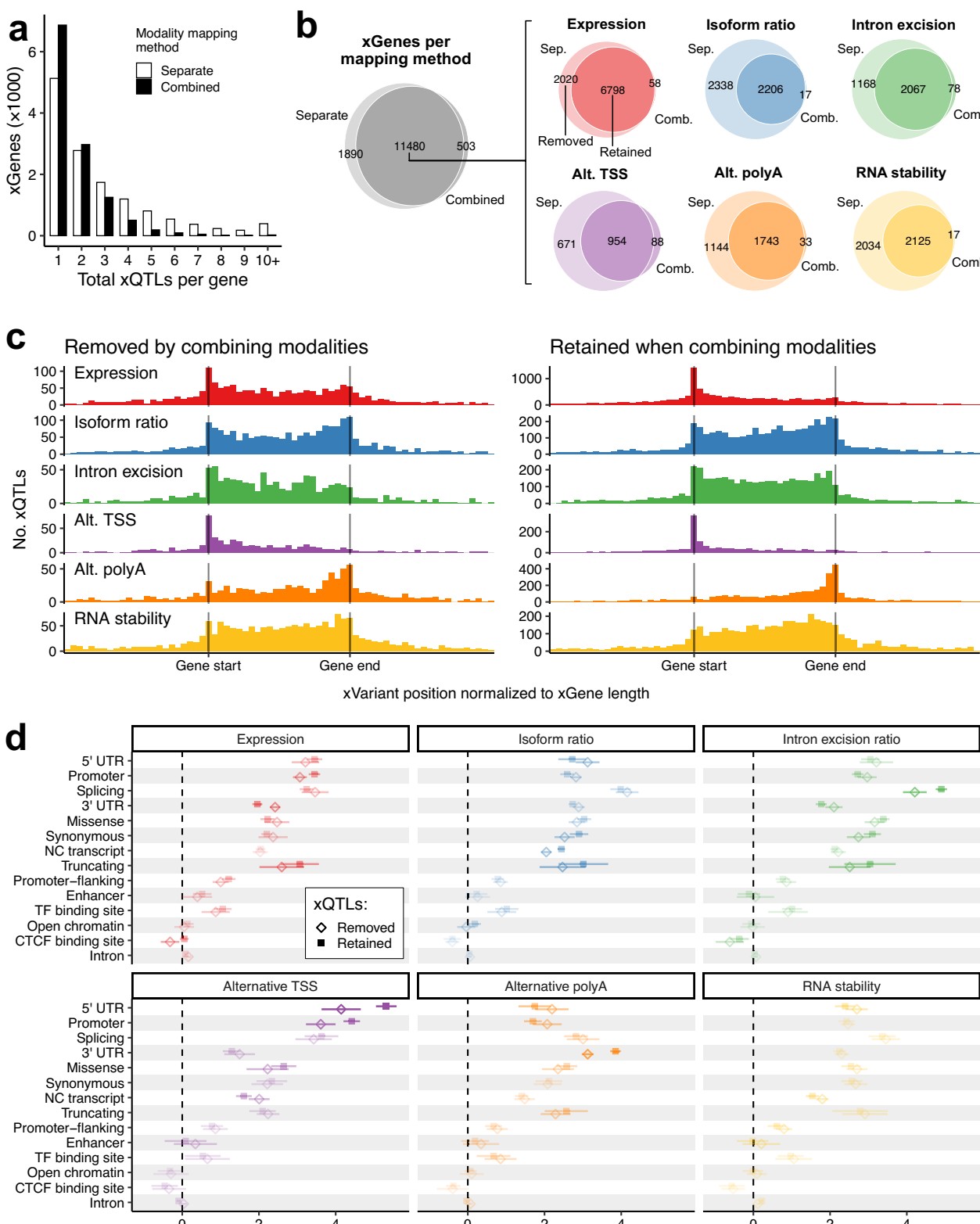

**Fig. 3 | Comparison of separate-modality and cross-modality xQTL mapping. a** Total xQTLs per xGene in Geuvadis, summed over modalities, when testing for conditionally independent xQTLs separately per modality and when testing across all modalities. **b** Left: Venn Diagram for xGenes from separate-modality mapping and from cross-modality mapping in Geuvadis. Right: For the genes with xQTLs from both mapping methods, the same comparison for the subsets of xGenes for each modality. Colors represent modalities. **c** The same type of relative xQTL position plots as Fig. 2d, but for the two subsets of xGenes per modality, removed and retained, indicated in (**b**). Histograms are colored by modality. **d** Enrichment of functional annotations in the xVariants in 49 GTEx tissues, similar to Fig. 2e but for the removed and retained xGene subsets determined for each tissue. Opacity of the points and segments is proportional to the distance between each pair of points. Points and bars are colored by modality. Source data are provided as a Source Data file.

respectively. These observations indicate that cross-modality mapping results are more biologically plausible. We also compared functional annotation enrichment in removed and retained xQTLs and observed similar characteristic differences (Fig. 3d). These include stronger enrichment of promoter annotations and weaker enrichment of 3' UTR annotations in expression QTLs; stronger enrichment of splicing annotations in intron excision ratio QTLs; stronger enrichment of 5' UTR and promoter annotations in alternative TSS QTLs and the opposite in alternative polyA QTLs; and stronger enrichment of 3' UTR annotations in alternative polyA QTLs.

**xTWAS doubles the discovery of trait-associated genes.** While TWAS is most commonly applied to gene expression data, the underlying principles and models are largely applicable to any of the RNA phenotypes provided by Pantry. We trained TWAS models on all RNA phenotypes (xTWAS), training one model per phenotype in the same way as the conventional method of training one expression model per gene. For modalities with multiple RNA phenotypes per gene, this produced multiple models per gene. We performed xTWAS on a published collection of harmonized data for 114 traits, including cardiometabolic, psychiatric-neurologic, anthropometric, immune, blood, and other trait categories[14]. For Geuvadis, we found 10,065 significant hits across 80 traits involving 4,304 unique RNA phenotypes for 1934 genes. Of the 4,487 unique trait-gene pairs among these hits, 51.3% involved only non-expression RNA phenotypes, and thus would not have been identified in a typical expression-only TWAS analysis (Fig. 4a). While xTWAS produced a dramatic increase in findings compared to TWAS, expression phenotypes produced the single largest number of TWAS hits, and the most top hits per gene, of any modality.

For each xTWAS hit, we sought more stringent evidence for mediation by testing for colocalization of the RNA phenotype and trait genetic associations using COLOC[15], which is a more conservative test than TWAS[14]. Approximately one-third of the xTWAS hits exhibited strong evidence of colocalization at a shared variant (posterior probability of association >0.8), ranging from 32.1% to 36.3% per modality (Fig. 4b). That is, no modality was especially depleted of colocalizations among its TWAS associations. We also ran xTWAS on each GTEx tissue (Supplementary Data 3), identifying colocalizing hits for 50,442 more trait-tissue-gene triplets than would be found using expression alone, a 2.73-fold change (Fig. 4c). This effect occurred across all tissues, while the total number of colocalizing hits per tissue varied widely due to factors such as sample size. Aside from several tissues with very low sample size, the proportion of TWAS hits per tissue-modality pair with strong evidence of colocalization was similar to that observed for Geuvadis (Supplementary Fig. 6).

To measure concordance of xTWAS associations between independent datasets, we compared results from Geuvadis to those from GTEx LCL. There were 2179 trait-gene pairs significant in both Geuvadis and GTEx LCL, which is 49% of the 4487 Geuvadis (sample size = 445) significant pairs and 63% of the 3450 GTEx LCL (sample size = 147) significant pairs.

Since RNA phenotypes were individually tested for xTWAS associations, we also performed association testing using FOCUS, a method that uses a fine-mapping approach to handle the confounding effects of linkage disequilibrium (LD) and pleiotropy[16]. We ran FOCUS using FUSION's transcriptomic models for all Pantry modalities trained on Geuvadis data. We observed similar proportions of the modalities among the top associations per trait-gene pair as compared to those from FUSION, though the expression proportion was higher, at 41.6% compared to 35.6% for FUSION hits (Supplementary Fig. 7). While the FOCUS analysis required modification to accommodate Pantry's multimodal phenotypes (see Methods), these results suggest that the observed contributions of each RNA modality to TWAS discovery are not strongly affected by confounding due to LD, co-regulation, or pleiotropy.

**xTWAS enhances understanding of GWAS results.** GWAS loci are often provisionally attributed to the nearest gene, although it is generally understood that the nearest gene may or may not have a mediating role. We identified the two nearest genes to each GWAS locus for all traits and matched those trait-gene pairs with Geuvadis colocalizing xTWAS hits. Across the 7071 loci, 566 (8%) could be potentially explained by an xTWAS hit matching one of the two nearest genes. Of those loci, 333 (59%) matched only non-expression hits. We repeated this analysis with colocalizing xTWAS hits from all 49 GTEx tissues after applying a more stringent Bonferroni threshold for TWAS p-values that accounts for the number of tissues in addition to the number of modalities. We found that 1906 loci (27%) could be potentially explained by a hit in any tissue. Of those, 651 (34%) matched only non-expression hits. Compared to the single-tissue Geuvadis data, xTWAS hits across 49 tissues provided more contexts in which to detect potential mediators. While this resulted in more loci having at least one matching expression hit, this multi-tissue analysis still resulted in 95% more loci potentially explained by exclusively non-expression colocalizing xTWAS hits.

We also examined GTEx xTWAS hits for which the gene was originally reported in the GWAS study as a potential mediator based on its proximity to an association locus. For example, from the colocalizing xTWAS hits for neuroticism in UK Biobank, we identified several genes known previously to be relevant to behavior: *ORC4*, *CRHR1*, and *DRD2*. All three were reported in a GWAS on neuroticism in UK Biobank as being within associated regions[17]. In our xTWAS, their associated modalities included only isoform ratio and/or intron excision ratio across all 13 brain tissues for *ORC4* (Fig. 5); expression, isoform ratio, intron excision ratio, alternative TSS, and/or RNA stability in four brain tissues for *CRHR1*; and expression in one brain tissue, cerebellum, for *DRD2*.

We also examined biologically relevant genes that had been reported based on colocalizing eQTLs rather than proximity alone. For example, in the PGC schizophrenia GWAS, *CYP2D6*, which encodes a pharmacologically important P450 enzyme[18], was included among blood eQTLs, but not brain eQTLs, that fell within a GWAS locus credible set[19]. We found colocalizing xTWAS hits for this schizophrenia trait for *CYP2D6* in five GTEx brain tissues, liver, and seven other tissues, all for isoform ratio, intron excision ratio, or RNA stability phenotypes, and none for expression phenotypes.

A GWAS for sleep duration in UK Biobank found *PER1*, a well-characterized circadian rhythm gene, within an associated locus[20]. We found colocalizing xTWAS hits for circadian rhythm for *PER1* in thyroid, coronary artery, and sigmoid colon, all of which were for the alternative TSS modality. For a related trait, morning/evening person chronotype, a GWAS in UK Biobank followed by pathway analysis identified *RELN*, a gene previously linked to schizophrenia but not circadian rhythm[21]. We found colocalizing xTWAS hits for *RELN* exclusively in cerebellar hemisphere, cerebellum, and tibial nerve, for morning/evening person chronotype, and for alternative polyA modality, and did not find any other hits for other tissues, traits, or modalities.

**RNA modalities harbor largely consistent proportions of genetic regulation across tissues.** We examined the proportion of xQTLs and xTWAS hits coming from each RNA modality for each tissue to identify trends or outliers that could have biological significance. The proportions were fairly consistent across GTEx tissues, with no strong relationship to sample size (Supplementary Fig. 8). A notable deviation was in Testis, which had the highest proportion of intron excision ratio phenotypes in both xQTLs (28.3%) and xTWAS hits (29.1%) of any tissue. This observation is consistent with existing knowledge that alternative splicing is especially prevalent in the testis[22]. Another strong deviation was cultured fibroblasts having a relatively high fraction of xQTL hits for RNA stability (13.4%, compared to mean 8.7% across tissues).

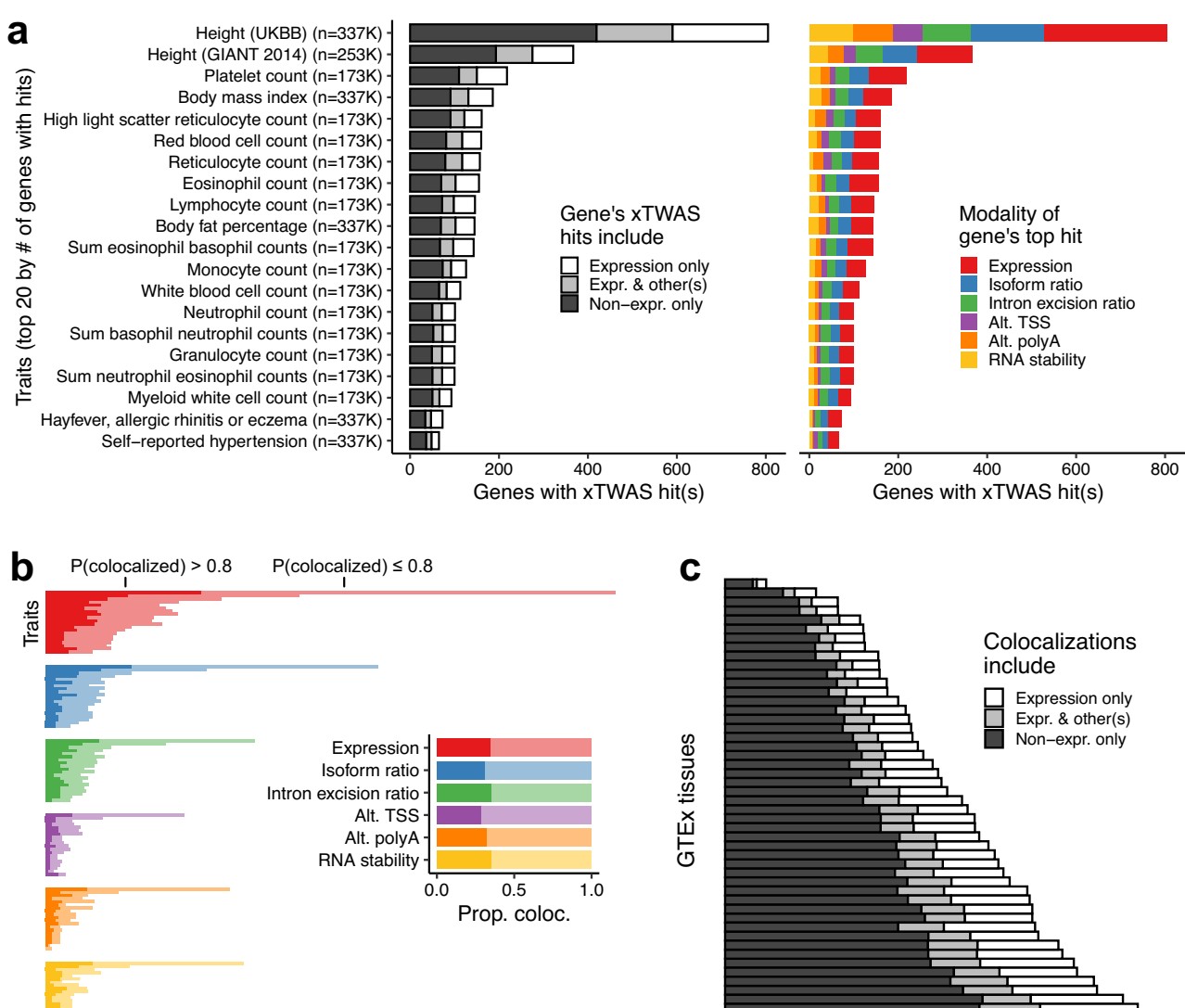

**Fig. 4 | Multimodal TWAS (xTWAS). a** For each trait, the number of genes with at least one xTWAS hit using Geuvadis RNA phenotypes are shown, shaded by whether each gene's hits(s) were for its expression phenotype, one or more other RNA phenotypes, or both. Only the top 20 traits in terms of gene count are shown[44–46]. xTWAS hits are associations with TWAS *p*-values reported by FUSION below a genome-wide *p*-value threshold, Bonferroni adjusted for the number of RNA modalities, of $8.33 \times 10^{-9}$. To the right, the same traits and genes are colored by the modality of each gene's hit with the lowest xTWAS *p*-value. In both plots, each gene is represented at most once per trait, and genes in the "Expression only" category on the left overlap with, but are not the same set as, genes in the "Expression" top hit category on the right. UKBB, UK Biobank; GIANT, Genetic Investigation of Anthropometric Traits. **b** For the same 20 traits, the number of TWAS hits per modality which also have strong evidence of single-variant-level colocalization is indicated with shading. The inset shows mean colocalizing proportions per modality. Probabilities are posterior probabilities, computed by COLOC, of a shared causal variant underlying a given TWAS association. **c** Colocalized gene-trait pairs for each GTEx tissue, shaded by whether the colocalization(s) involved an expression phenotype, one or more other RNA phenotypes, or both. Source data are provided as a Source Data file.

## Discussion

We have introduced Pantry, a framework for multimodal analysis of RNA-seq data and its application to xQTL discovery and GWAS interpretation. Pantry dramatically increases the number of genomic discoveries when used to reanalyze previously generated datasets. Notably, for more than two-fifths of the gene-trait pairs with previous TWAS hits from gene expression analysis, we identified at least one additional regulation modality. While these genes are not completely new discoveries, the association with the new modality may facilitate the identification of the biological mechanism driving the association. We have shown that the systematic analysis of multiple RNA modalities reveals complementary biological information and genetic signals, improving the number and the specificity of genetic discoveries as compared to the conventional gene expression-based analysis using the same data. Finally, we share all the tools, methods and generated data with the community, including the RNA phenotypes, xQTLs, xTWAS weights, and xTWAS associations generated from the GTEx project and Geuvadis data.

The Pantry framework is modular and amenable to the addition of other transcriptomic modalities not considered here to facilitate further expansion and adaptation by the genomics community. These could include types of RNAs lacking polyA tails, which may only be

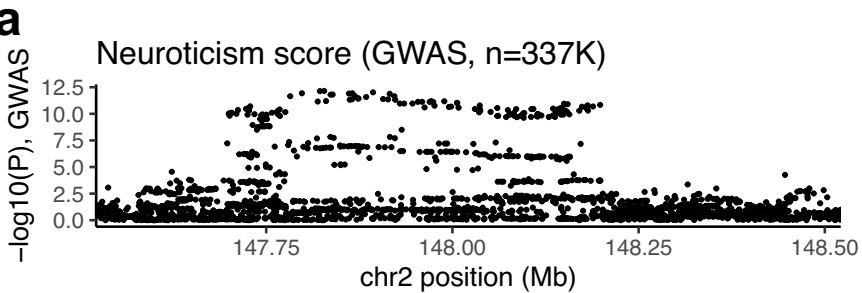

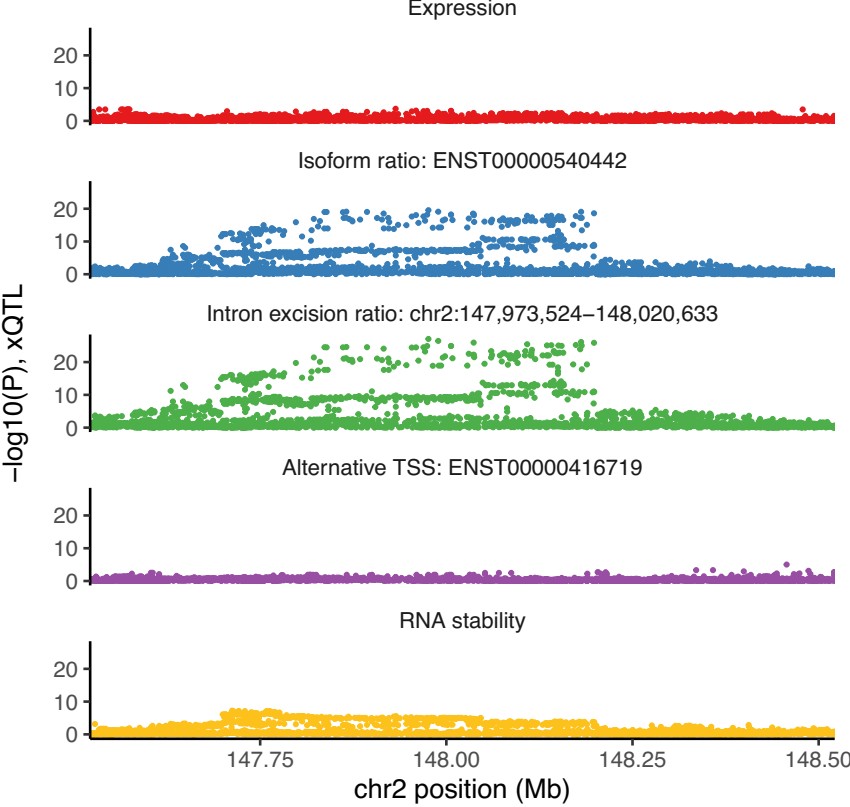

**Fig. 5 | Example of a non-expression xTWAS hit. a** GWAS *p*-values from a linear SNP association model for the neuroticism trait within the *ORC4* cis-window on Chromosome 2, the region for which we performed xTWAS. **b** The *p*-values from xQTL mapping in GTEx cortex tissue (BRNCTXA, *n* = 205 samples), for four *ORC4* RNA phenotypes, in the same Chromosome 2 interval. Significant xTWAS hits for neuroticism were only found for isoform ratio and intron excision ratio modalities. Only the *ORC4* phenotype with the lowest cis-QTL *p*-value per modality is shown. Unadjusted nominal *p*-values produced by tensorQTL for Pearson correlations between normalized residual genotypes and phenotypes are shown. Source data are provided as a Source Data file.

sufficiently quantified in non-polyA-selected RNA-seq libraries. Alternative forms of existing modalities, such as different ways to represent isoform abundance or more abstract features that represent expression variation, could be explored using this framework.

The established technique of using stepwise regression to find conditionally independent xQTLs naturally lends itself to multi-modal RNA phenotype data. Not only can it avoid redundant xQTLs in the presence of multiple phenotypes of the same modality, such as those representing alternative splice junctions, but it also avoids redundant xQTLs across modalities. However, when two phenotypes of different modalities share an xQTL signal, one modality might reflect the underlying causal mechanism better than the other, and that phenotype may not always have the stronger association. Thus, while the multi-modal conditionally independent xQTL mapping does sharpen

the modality-specific functional characteristics of the xQTLs overall when compared to separate-modality mapping, it may also remove some of the true associations to a lower powered transcriptional modality (e.g., RNA stability) in favor of a better powered one (e.g., gene expression). Furthermore, a regulatory variant could lead to two distinct effects on different modalities, for example by altering splice junction usage, which increases the rate of nonsense-mediated decay (NMD), which changes total expression level[23]. We therefore also provide results from each modality individually analyzed for applications that benefit from more comprehensive sets of modalities and to resolve causal chains of distinct molecular effects, as well as applications focused on a single modality of gene regulation.

The abundance of all six modalities among the cross-modality mapped xQTLs highlights the benefit of quantifying many modalities,

even those that overlap in the transcriptomic variation they represent. For example, isoform ratios and intron excision ratios capture alternative splicing variation in different forms, and they are roughly equally represented among the xQTLs selected to represent conditionally independent genetic signals. This is consistent with the observation of Garrido-Martín et al.[4] that the two approaches are complementary, one having higher power to detect sQTLs than the other in specific cases, depending on properties of the gene. They also note that isoform ratios can represent additional types of variation, such as alternative TSS and polyA sites, not captured by intron excision ratios.

There are existing methods such as isoTWAS[24] and OPERA[5] that incorporate molecular information beyond total gene expression into genetic analysis[25]. Such methods have demonstrated that many more gene-trait associations can be discovered compared to using gene expression alone. Other studies have shown that cis-regulatory variants can be mapped for various transcriptional modalities beyond gene expression[1,3,7,9,26,27]. Pantry's strength is in providing a framework that begins with the raw RNA-seq data, produces comprehensive transcriptional phenotypes, and applies them seamlessly to multimodal genetic analyses. While Pantry currently tests RNA phenotypes individually for TWAS associations, a method that jointly models multiple phenotypes has the potential for more powerful association testing. However, such a method would need to handle the complexity of multimodal RNA phenotypes, either by working with potential differences in measurement error across the modalities, or by testing each modality separately such that the results can be interpreted in a comparable way across modalities. Further research is needed to develop joint multimodal xTWAS methods for integration into Pantry.

This study has several important limitations. Pantry would require modification to handle single cell RNA-seq data. The power afforded by the GTEx tissue sample sizes leave some observable genetic associations undetected, particularly for the brain tissues where GTEx sample sizes are smaller. RNA-seq datasets such as those analyzed here may not cover the developmental stage, environmental exposures, or ancestry groups in which a transcriptomic mediator would be active and detectable. Still other molecular mediation could be only detectable in other types of omics data, such as DNA methylation or proteins. For species with sparser reference transcriptome data, Pantry would produce fewer RNA phenotypes, leading to fewer discovered genetic associations. Finally, xQTL mapping and xTWAS primarily detect associations for common variants; other techniques would need to be employed to detect regulatory effects of rare or de novo variants[28].

We have reported both broad characteristics of the xQTL and xTWAS results and specific observations that demonstrate Pantry's utility. However, given the high dimensionality of these analyses (tissues, genes, modalities, and often multiple phenotypes per modality for xQTLs, and the additional dimension of traits for xTWAS), we expect many more interesting biological insights to be found in the data repository published alongside this study. Furthermore, the inclusion of intermediate data such as the RNA phenotype quantifications and TWAS models for all GTEx tissues can enhance future methods development and GWAS. Use of the Pantry code on additional transcriptomic datasets, including those with large cohorts such as ROSMAP[29] and PsychENCODE[30], will provide deep and comprehensive genetic analyses of the transcriptome.

## Methods

### Geuvadis dataset
We downloaded the quality control-filtered Geuvadis RNA-seq dataset ($n = 445$ lymphoblastoid cell line samples) and corresponding genotypes for 13.4 million variants. These were filtered to autosomal biallelic variants with minor allele frequency (MAF) >= 0.01, resulting in 12.9 million variants. We ran the data through the default Pantry phenotyping and Pheast pipelines.

### GTEx datasets
We downloaded the RNA-seq reads for all 54 GTEx v8 tissues. We obtained corresponding genotypes for 10.7 million variants and filtered to autosomal biallelic variants with MAF >= 0.01, resulting in 10.4 million variants. We ran the data through the default Pantry and pipeline, and ran QTL and TWAS analyses on the 49 tissues originally selected for eQTL mapping in GTEx v8.

### RNA phenotyping
RNA phenotypes were generated using default Pantry parameters. We used human genome reference version GRCh38 and version 106 Ensembl gene annotations. Genes were not filtered by their annotated biotype, but final processing of results included filtering to protein-coding and lncRNA genes for statistics and visualizations. For RNA stability phenotypes, we filtered annotations to those with the Ensembl pipeline as an annotation source to limit rare or speculative isoforms that would prevent any constitutive exons from being counted for many genes. Pantry uses a two-stage quantile normalization procedure for RNA phenotype tables as implemented in pyQTL [https://github.com/broadinstitute/pyqtl/blob/master/qtl/norm.py]. First, samples are quantile normalized such that all samples have the average empirical distribution, to control for transcriptome-wide distribution effects, e.g., those caused by variation in highly expressed genes. Then, each phenotype is inverse-normal transformed (preserving ties) to prevent issues with linear regression caused by unusual distributions.

### Covariates
We used code included in the Pantry Pheast module to compute covariates. For each modality in each dataset (tissue), we ran principal component analysis (PCA) on the RNA phenotype table and used the first 20 principal components (PCs) as covariates. We also ran PCA on each LD-pruned genotype alternative allele count matrix and included the first 5 PCs as covariates. We tested the impact of also including sex or age from the metadata as covariates, but found little impact on xQTL results (Supplementary Fig. 9). Since a covariate strategy that only uses data-derived PCs reduces the complexity of preparing new datasets for Pantry, we made it the default for Pantry and used it for the analyses in this study.

### xQTL mapping
We mapped conditionally-independent cis-QTLs for each modality in each dataset (tissue) using tensorQTL[12], running the default commands included in Pantry Pheast. Modalities with multiple phenotypes per gene were mapped as per-gene groups so that cis-QTLs were conditionally independent across phenotypes within each gene.

For cross-modality mapping, the RNA phenotype tables per dataset (tissue) were concatenated, and 25 total covariates were computed in the same way as for individual modalities using the combined phenotype table and the genotypes. We then mapped cis-QTLs for this combined dataset, grouping all phenotypes per gene so that cis-QTLs were conditionally independent across all phenotypes of all modalities within each gene.

### Allelic fold change validation using allele-specific expression
To compare the robustness of cis-eQTLs from different mapping strategies, we measured the effect size of cis-eQTLs in GTEx subcutaneous adipose tissue (ADPSBQ), both for eQTLs found when mapping modalities separately (SMM-eQTLs) and those found from cross-modality mapping (CMM-eQTLs). We estimated aFC using the aFC-n model[31] with phased genotypes. We also calculated aFC from allele-specific expression in heterozygous individuals using phASER[32].

To avoid potential confounding effects of different stringencies of eQTLs, we matched *p*-value distributions between the eQTL sets. Specifically, we removed CMM-eQTLs with *p*-values higher than the maximum SMM-eQTL *p*-value, and then found the SMM-eQTL with the closest log(*p*-value) to that of each CMM-eQTL. This resampling of SMM-eQTLs resulted in the two sets of eQTLs having the same size and nearly the same *p*-value distribution, and we plotted and measured aFC agreement in these two sets.

## xTWAS

We downloaded summary statistics from a collection of 114 GWAS traits[14] from Zenodo record 3629742 [https://doi.org/10.5281/zenodo.3629742][33]. We ran TWAS analysis using FUSION[34] with default parameters. First, we fit predictive models for each Geuvadis RNA phenotype using TSS ± 500 kb cis-window genotypes, along with the same 25 covariates used for cis-QTL mapping, and ran FUSION's built-in comparison of blup, lasso, top1, and enet models for each phenotype. We then used these models and the recommended LD reference data from 1000 Genomes to test TWAS associations for each RNA phenotype against each GWAS trait. We used a genome-wide *p*-value threshold, Bonferroni adjusted for the number of RNA modalities, of $8.33 \times 10^{-9}$ ($5 \times 10^{-8}$ divided by 6) to determine significant TWAS hits. We also used FUSION's built-in option to report COLOC posterior probabilities for each hit.

For the GWAS loci-based analysis, we determined loci for each of the 114 traits by extracting all genome-wide significant ($P < 5 \times 10^{-8}$) variants and grouping them such that any two significant variants <500 Kb apart were in the same locus. We found the two nearest genes to each locus based on the distance between the variant in the locus with the lowest *p*-value and the nearest point in the gene's interval. For each locus, xTWAS hits matching the trait and either of the two nearest genes were assigned to the locus. For this analysis, we used only colocalizing xTWAS hits (those with COLOC posterior probability of association >0.8), and for GTEx hits, used a more stringent TWAS *p*-value threshold that was Bonferroni adjusted for the number of tissues in addition to the number of RNA modalities, i.e., $1.70 \times 10^{-10}$.

## FOCUS for validation of xTWAS

We prepared a transcriptomic weights database for FOCUS by including the Geuvadis FUSION models for all six Pantry modalities. Since FOCUS is not designed to handle multiple transcriptomic phenotypes per gene per tissue, using actual gene IDs resulted in combinatorial explosion and insurmountable out-of-memory errors. Instead, we treated each RNA phenotype as its own gene. For each of the 114 GWAS traits, we ran "focus finemap" on this database, GWAS summary stats, the same LD reference as used for FUSION, and "--locations 38:EUR". We recovered the actual gene IDs for each result and used posterior inclusion probability >0.8 to quantify RNA phenotype-trait associations.

## Variant effect enrichment

We downloaded variant annotations from the GTEx Portal [https://storage.googleapis.com/adult-gtex/references/v8/reference-tables/WGS_Feature_overlap_collapsed_VEP_short_4torus.MAF01.txt.gz]. To reduce low-frequency annotation categories, we merged Splice acceptor, Splice donor, and Splice region categories into one Splicing category, and merged Frameshift and Stop gained into one Truncating category. For the conditionally independent cis-QTLs for each RNA modality for each GTEx tissue, enrichment of each annotation in the xVariants was computed as the $\log_2$-ratio of the proportion among the xVariants to the proportion in all variants within all the cis-windows tested for that RNA modality. To control the variance of enrichment values from infrequent annotations that result in low annotated xVariant counts, we added a pseudocount of

0.5 to each annotated xVariant count, and added an amount to the total xVariant counts such that the added xVariants had background annotation frequency. We also omitted tissue-modality-annotation combinations with fewer than 2 annotated xVariants from enrichment analysis.

## Reporting summary

Further information on research design is available in the Nature Portfolio Reporting Summary linked to this article.

## Data availability

The data processed with Pantry for Geuvadis and all GTEx tissues are available at https://pantry.pejlab.org and in a public Zenodo repository [https://doi.org/10.5281/zenodo.13922139][35]. These include, for all six modalities in each tissue, RNA phenotype matrices, covariates, xQTLs, xTWAS transcriptomic model weights, and xTWAS associations for 114 GWAS traits. This repository is about 42 GB when compressed. Raw Geuvadis data were downloaded from ArrayExpress, accession E-GEUV-1. Raw protected GTEx data were downloaded from the database of Genotypes and Phenotypes (dbGaP), accession no. phs000424.v8 [https://www.ncbi.nlm.nih.gov/projects/gap/cgi-bin/study.cgi?study_id=phs000424.v8.p2]. GWAS summary statistics used for xTWAS are available at Zenodo record 3629742 [https://doi.org/10.5281/zenodo.3629742][33]. Source data are provided with this paper.

## Code availability

The Pantry code is maintained at https://github.com/PejLab/Pantry with version 1.0.0 used for this study[36]. It is structured as a two-stage pipeline using the Snakemake workflow management system[37]. The pipeline consists of existing programs, e.g., STAR[38] and samtools[39], and additional scripts to process their input and output data. The pipeline was designed for computational and storage efficiency by reducing redundant computation and large files, and is compatible with high performance computing environments. See Supplementary Methods for details.

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

## Acknowledgements

We thank Robert Vogel for helpful discussions. Funding: National Institute on Drug Abuse [P50DA037844, A.A.P.], National Institute of General Medical Sciences [R01GM140287, P.M.], National Institute of Mental Health [R01 MH125252, A.G.].

## Author contributions

D.M. wrote the code and processed, analyzed, and visualized the data. N.E. and S.M.E. tested and debugged the code. A.G., A.A.P., and P.M. supervised the project. D.M., A.G., A.A.P., and P.M. wrote the manuscript. All authors edited and approved the manuscript.

## Competing interests

The authors declare no competing interests.
