## [Peer Review File · Nature Communications]

REVIEWER COMMENTS

Reviewer #1 (Remarks to the Author):

The research paper presents the Pantry framework, which enables the generation of diverse RNA phenotypes and their integration with genetic data for comprehensive analyses. The Pantry framework introduces a novel approach for analyzing RNA-seq data, facilitating the quantification of multiple RNA modalities and their integration with genetic data for downstream analyses. Demonstrations of Pantry's application to 50 human tissue datasets resulted in a significant increase in significant gene-trait associations and improved specification of relevant RNA phenotypes through xTWAS. Utilization of cross-modality mapping efficiently mapped cis-QTLs for combined datasets, leading to fewer total xQTLs per gene on average and enhancing the specificity of genetic discoveries.

The transparency and reproducibility of the study, evidenced by the availability of Pantry code, RNA phenotypes, xQTLs, and xTWAS results for community access, fosters further research and method development in genomics. Overall, the research contributes significantly to the field of complex trait genetics by providing a comprehensive framework for multimodal analysis of RNA sequencing data, making it a valuable addition to the scientific literature.

Despite these advancements, challenges such as handling single-cell RNA-seq data and addressing practical issues like data formatting and computational resources remain. Future directions may include expanding the analysis to include additional RNA modalities, diverse tissues, and datasets to improve the generalizability of the findings.

Reviewer #2 (Remarks to the Author):

The authors build and test a framework, Pantry, for cross modality QTL mapping and downstream analysis, focusing on the various modalities that can be quantified from short read bulk RNA-seq. They apply Pantry to Geuvadis and GTEx, evaluating the number of detected xQTLs for each modalities, the effect of grouped mapping, and enrichment in specific genomic annotations. A cross-modality TWAS ("xTWAS") and coloc is then conducted using the same set of trait GWAS as the GTEx TWAS/coloc paper, with the main conclusion being that many trait-gene pairs are missed if one only looks at total expression.

The paper is clearly written and well motivated. The method could use a little more detail:

- The grouped phenotype stepwise regression approach of tensorQTL should be summarized in the methods since this is key to a lot of Pantry's analysis.
- what normalization is performed? e.g. quantile normalization? Motivate these choices.
- are permutations used in tensorQTL? Hopefully yes since I don't believe stepwise regression can produce valid p-values.
- for some of the figs it would be helpful to see (effective) sample sizes (for the GWAS traits for example when looking at multiple or for the GTEx tissues when comparing them).

They make the argument that grouped analysis is finding the more "real"/"plausible" QTLs. I think reality is more complex than this. For example, a variant may directly effect the splicing of a gene, e.g. by weakening a splice site. That may (in fact quite often according to Yang Li's recent work) result in NMD and reduced expression. The reduced expression may then be the direct cause of trait variation, but splicing was the direct effect of the variant. So the full chain is SNP -> splicing -> expression -> trait. The paper doesn't try to address causality but that typically is what we care about, and in this case the SNP does cause expression change (just indirectly), and simply splicing does cause the trait (again indirectly). This should be discussed at least.

"Cross-modality mapping improves quality and interpretation of xQTLs": it is possible the grouped analysis is simply more conservative. The authors should adjust the non-grouped significance threshold to give a similar number of discoveries to the grouped analysis, and see if this equally improves the aFC agreement.

Does FUSION deal with co-regulation somehow now? e.g. along the lines of FOCUS? If not, FOCUS (or equivalent) should also be run and the results compared.

The traditional thinking was that isoform ratio QTLs were underpowered compared to QTLs for local splicing quantification (leafcutter etc). The authors don't see that - is this due to changes in methodology? Improvements in quantification using kallisto?

I'm surprised tissue sample size doesn't effect the TWAS/coloc results - can the authors comment on this? Brain in particular has small sample size in GTEx and this is unfortunate given the focus on neuro traits and splicing. Ideally the authors would additionally analyze a larger brain cohort such as ROSMAP.

Overall the work is thorough and well presented.

Reviewer #2 (Remarks on code availability):

I've only reviewed this briefly but it looks well documented and maintained.

Reviewer #3 (Remarks to the Author):

In this well-written manuscript, Munro et al. present a suite of computational tools (Pantry/Pheast) for the generation and genetic analysis of multiple RNA transcriptional modalities. The authors apply their tools to generate multimodal data from Geuvadis and GTEx. Subsequent QTL mapping of the resulting data in these two studies identified a large collection of genes possessing variants with significant effect on modalities distinct from total gene expression. The authors also used the resulting multimodal data to perform TWAS on over 100 complex human traits and showed the use of multimodal data greatly increased gene discovery over the use of total gene expression alone; thereby suggesting that the study of additional modalities can elucidate the origins of human traits and diseases. The paper has many strengths. The Pantry framework likely will have substantial value in a variety of existing studies of gene regulation (with the new multimodal data generated for such studies also having value in related projects like TWAS). I also appreciated the authors making the Geuvadis/GTEx multimodal data available (along with the QTL and TWAS results) to facilitate further analyses based on these popular datasets.

Main Comments:

The methodology underlying the xQTL mapping procedure requires quite a bit more detail as it is difficult to evaluate. Specifically, it is unclear how such mapping is performed when considering multiple phenotypes (either the cross-modality procedure or the analysis of a specific modality with multiple phenotypes per gene) and how the correlation among related phenotypes are considered within analysis. The manuscript mentions their mapping method is based on tensorQTL, which I believe is a GPU implementation of FastQTL that is designed for a single phenotype (rather than multiple). A supplemental section providing the technical details of model fitting would be helpful.

Minor Comments:

Lines 147-148: For the concordance analysis, what proportion of variant-RNA phenotype pairs were eligible to be tested for each phenotype in GTEx LCL data?

Lines 219-221: For TWAS analysis assuming a specific modality with multiple phenotypes, was the model trained separately for each phenotype? Would a training model that leveraged multiple phenotypes from the modality (in the spirit of Feng et al. PLoS Genet 17(4): e1008973 or other related methods) perhaps be more powerful?

Lines 225-227: Of the 4,487 trait-gene pairs that were significant using TWAS trained on Geuvadis, what percentage were also significant using TWAS trained on GTEx LCL data?

Figure 4C: Was any modality depleted/enriched for colocalization of TWAS associations in any of the GTEx tissues?

Lines 396-399: Why not adjust for gender in the xQTL and xTWAS analyses?

Response to reviewers

Reviewer #1 (Remarks to the Author):

The research paper presents the Pantry framework, which enables the generation of diverse RNA phenotypes and their integration with genetic data for comprehensive analyses. The Pantry framework introduces a novel approach for analyzing RNA-seq data, facilitating the quantification of multiple RNA modalities and their integration with genetic data for downstream analyses. Demonstrations of Pantry's application to 50 human tissue datasets resulted in a significant increase in significant gene-trait associations and improved specification of relevant RNA phenotypes through xTWAS. Utilization of cross-modality mapping efficiently mapped cis-QTLs for combined datasets, leading to fewer total xQTLs per gene on average and enhancing the specificity of genetic discoveries.

The transparency and reproducibility of the study, evidenced by the availability of Pantry code, RNA phenotypes, xQTLs, and xTWAS results for community access, fosters further research and method development in genomics. Overall, the research contributes significantly to the field of complex trait genetics by providing a comprehensive framework for multimodal analysis of RNA sequencing data, making it a valuable addition to the scientific literature.

Despite these advancements, challenges such as handling single-cell RNA-seq data and addressing practical issues like data formatting and computational resources remain. Future directions may include expanding the analysis to include additional RNA modalities, diverse tissues, and datasets to improve the generalizability of the findings.

Thank you for the remarks. We have expanded the discussion of the limitations of this study's analyses and of the current Pantry implementation (our additions in red):

“This study has several important limitations. Pantry would require modification to handle single cell RNA-seq data. The power afforded by the GTEx tissue sample sizes leave some observable genetic associations undetected, particularly for the brain tissues where GTEx sample sizes are smaller. RNA-seq datasets such as those analyzed here may not cover the developmental stage, environmental exposures, or ancestry groups in which a transcriptomic mediator would be active and detectable. Still other molecular mediation could be only detectable in other types of omics data, such as DNA methylation or proteins. For species with sparser reference transcriptome data, Pantry would produce fewer RNA phenotypes, leading to fewer discovered genetic associations. Finally, xQTL mapping and xTWAS primarily detect associations for common variants; other techniques would need to be employed to detect regulatory effects of rare or *de novo* variants.”

And:

“Use of the Pantry code on additional transcriptomic datasets, including those with large cohorts such as ROSMAP and PsychENCODE, will provide deep and comprehensive genetic analyses of the transcriptome.”

Reviewer #2 (Remarks to the Author):

The authors build and test a framework, Pantry, for cross modality QTL mapping and downstream analysis, focusing on the various modalities that can be quantified from short read bulk RNA-seq. They apply Pantry to Geuvadis and GTEx, evaluating the number of detected xQTLs for each modalities, the effect of grouped mapping, and enrichment in specific genomic annotations. An cross-modality TWAS ("xTWAS") and coloc is then conducted using the same set of trait GWAS as the GTEx TWAS/coloc paper, with the main conclusion being that many trait-gene pairs are missed if one only looks at total expression.

The paper is clearly written and well motivated. The method could use a little more detail:
- The grouped phenotype stepwise regression approach of tensorQTL should be summarized in the methods since this is key to a lot of Pantry's analysis.

This is a good point given the importance of the grouped phenotype stepwise regression procedure to Pantry's multimodal integration. We have added a section to the Supplementary Information called "Cross-modality mapping with tensorQTL" describing our use of tensorQTL's grouped phenotype stepwise regression approach in more detail. It includes a link to the tensorQTL code repository (<https://github.com/broadinstitute/tensorqtl>).

- what normalization is performed? e.g. quantile normalization? Motivate these choices.

Thank you for pointing out this issue. We have added specific information on normalization strategy and motivation to the Methods (line 450):

“Pantry uses a two-stage quantile normalization procedure for RNA phenotype tables as implemented in pyQTL (<https://github.com/broadinstitute/pyqtl/blob/master/qtl/norm.py>). First, samples are quantile normalized such that all samples have the average empirical distribution, to control for transcriptome-wide distribution effects, e.g. those caused by variation in highly expressed genes. Then, each phenotype is inverse-normal transformed (preserving ties) to prevent issues with linear regression caused by unusual distributions.”

- are permutations used in tensorQTL? Hopefully yes since I don't believe stepwise regression can produce valid p-values.

Yes, tensorQTL uses permutations to produce p-values, both in single-pass and stepwise regression modes. We have added more description of the tensorQTL cis-QTL mapping procedure in the Results (line 103, reproduced below) and in the Supplementary Information (new subsection "Cross-modality mapping with tensorQTL"):

“Pantry Pheast combines the sets of phenotypes across all modalities and maps cis-QTLs with stepwise regression, considering all phenotypes per gene as a single group. This is implemented using tensorQTL’s stepwise regression, grouped phenotype, and data permutation features (Supplementary Information).”

- for some of the figs it would be helpful to see (effective) sample sizes (for the GWAS traits for example when looking at multiple or for the GTEx tissues when comparing them).

This was a good suggestion, we have now added GWAS sample sizes to Figure 4a and Figure 5, and added GTEx tissue sample sizes to Figure S2.

They make the argument that grouped analysis is finding the more "real"/"plausible" QTLs. I think reality is more complex than this. For example, a variant may directly effect the splicing of a gene, e.g. by weakening a splice site. That may (in fact quite often according to Yang Li's recent work) result in NMD and reduced expression. The reduced expression may then be the direct cause of trait variation, but splicing was the direct effect of the variant. So the full chain is SNP -> splicing -> expression -> trait. The paper doesn't try to address causality but that typically is what we care about, and in this case the SNP does cause expression change (just indirectly), and simply splicing does cause the trait (again indirectly). This should be discussed at least.

This is true, and the presentation and discussion of Pantry's cross-modality mapping should include the scenario in which correlated xQTLs for different modalities indeed reflect multiple effects along a causal chain. We have made two changes to the text to remedy this.

- We now mention this type of scenario in the Results (lines 101 and 191). We also acknowledge this issue as a limitation because Pantry cannot currently distinguish between the scenario in which a single type of transcriptomic variation appears as “redundant” genetic associations in multiple modalities, and the scenario of multiple distinct stages of true biological variation. Accordingly, we renamed this Results subsection from “Cross-modality mapping improves quality and interpretation of xQTLs” to “Cross-modality mapping reduces redundancy of xQTLs”.
- On line 376 we clarify the utility of the separate-modality mapped xQTL results that we provide alongside the cross-modality mapped xQTLs:
“Furthermore, a regulatory variant could lead to two distinct effects on different modalities, for example by altering splice junction usage, which increases the rate of nonsense-mediated decay (NMD), which changes total expression level [citation to Yang Li's recent work, “Global impact of unproductive splicing on human gene expression”]. We therefore also provide results from each modality individually analyzed for applications that benefit from more comprehensive sets of modalities and to resolve causal chains of distinct molecular effects, as well as applications focused on a single modality of gene regulation.”

"Cross-modality mapping improves quality and interpretation of xQTLs": it is possible the grouped analysis is simply more conservative. The authors should adjust the non-grouped

significance threshold to give a similar number of discoveries to the grouped analysis, and see if this equally improves the aFC agreement.

This is a good point, and we have updated this analysis and figure to reflect this consideration. The consolidating effect of cross-modality mapping could indeed be more likely to retain the stronger associations for a given modality than the weaker ones, which on its own could affect the aFC agreement. However, the median beta-approximated permutation p-values for separately mapped (SM) and combined-mapped (CM) eQTLs were similar ($6.6e-5$ and $6.1e-5$, respectively). If we removed the least-significant SM eQTLs to match the number of CM eQTLs, the median p-value for SM eQTLs would drop to $6.5e-8$. Since the p-values from the two mapping approaches represent the same underlying statistical tests, adjusting the significance threshold to match the number of eQTLs would lead to a much more stringent selection of SM eQTLs than of CM eQTLs.

We therefore decided to match p-value distributions between the eQTL sets for this analysis to equalize the number of eQTLs. Specifically, we removed CM eQTLs with p-values higher than the maximum SM eQTL p-value, and then found the SM eQTL with the closest $\log(p\text{-value})$ to that of each CM eQTL. This resampling of SM eQTLs resulted in the two sets of eQTLs having the same size and nearly identical p-value distributions. We have updated Figure S5, which still shows slightly higher aFC agreement when using the cross-modality mapping approach, and updated the associated text (line 184):

“Using data from GTEx subcutaneous adipose tissue, we measured allelic fold change (aFC) from gene expression data and again from allele-specific expression (ASE) data. These two measurements of cis-regulatory effect size are largely affected by independent sources of noise and as such allow us to gauge the quality of mapped cis-eQTLs. The Pearson correlation

between the two aFC measures was slightly higher for cross-modality mapping ($r = 0.721$, 95% CI [0.709, 0.733]) than for p-value-matched eQTLs from separate-modality mapping ($r = 0.703$, 95% CI [0.690, 0.715]), suggesting a refinement of eQTL signals (Supp Figure S5). We note, however, that in cases where a variant causes mechanistically distinct effects on multiple modalities, this cross-modality mapping procedure may also omit the xQTLs for some of those modalities.”

Does FUSION deal with co-regulation somehow now? e.g. along the lines of FOCUS? If not, FOCUS (or equivalent) should also be run and the results compared.

This is a good suggestion, and we have added an analysis and supplementary figure on FOCUS. Indeed, FUSION does not deal with co-regulation, so we tested for its potential confounding effects by running FOCUS using all Geuvadis FUSION models for each of the 114 traits. Since FOCUS is not designed to handle multiple transcriptomic phenotypes per gene per tissue, we were not able to successfully run it with multiple Pantry models per gene. Instead, we treated each RNA phenotype as a separate gene to run FOCUS and then quantified the results in terms of true genes. This resulted in 3,010 unique trait-gene pairs with posterior inclusion probability >0.8. For comparison, in our FUSION analysis there were 4,487 pairs with significant associations and 1,721 pairs with significant colocalizing associations. Looking at the top association for each of these trait-gene pairs, the proportion of modalities are similar across methods, though the expression proportion was higher for FOCUS, at 41.6% compared to 35.6% for FUSION hits. While the intended usage of FOCUS is not fully compatible with Pantry’s multimodal phenotypes, these results suggest that our conclusions regarding the contributions of each of Pantry’s modalities to TWAS discovery are not strongly affected by confounding due to co-regulation or pleiotropy. We show this in a new Supplementary Figure S7, referenced on line 278:

The traditional thinking was that isoform ratio QTLs were underpowered compared to QTLs for local splicing quantification (leafcutter etc). The authors don't see that - is this due to changes in methodology? Improvements in quantification using kallisto?

This is a good question, and development of the tools involved in quantification could indeed have affected the relative power of the two approaches over the years. In fact, Garrido-Martín et al. (2021 *Nature Communications*), when introducing their tool sQTLseekeR2, compared the quantification of transcript isoforms versus intron excision ratios from LeafCutter. While not identical to the isoform ratio and intron excision ratio phenotypes used in the present study, they found that the two approaches are complementary, one having higher power to detect sQTLs than the other in specific cases, depending on properties of the gene. They also note that isoform ratios can represent additional types of variation, such as alternative TSS and polyA sites, not captured by intron excision ratios. We have added text to the Discussion (line 383) on this pair of modalities and the findings of Garrido-Martín et al. in relation to those of the present study.

I'm surprised tissue sample size doesn't effect the TWAS/coloc results - can the authors comment on this? Brain in particular has small sample size in GTEx and this is unfortunate given the focus on neuro traits and splicing. Ideally the authors would additionally analyze a larger brain cohort such as ROSMAP.

We did not intend to suggest that tissue sample size did not affect TWAS/colocalization results. Figure S6 (now S8) does show largely consistent *proportions* of modalities among the TWAS hits, regardless of sample size, but as shown in Figure 4C and Table S3, the number of TWAS associations and colocalizations vary greatly across tissues, and sample size is likely a contributing factor. We have added this observation to line 261, and added Discussion text (line 409) on how the lower sample sizes for some GTEx tissues, particularly brain tissues, limited discoveries.

We agree with the reviewer that a larger brain dataset such as ROSMAP would provide a deeper analysis of regulatory variation and complex trait associations. However, we opted not to do this as performing a high quality and thorough analysis of this data will be beyond the scope of the present work. We believe the included data from the Geuvadis and GTEx cohorts are sufficient for the main focus of the work, which is to demonstrate the feasibility and utility of Pantry's multimodal transcriptomic and genetic analysis. That said, we do agree that a more focused, deep analysis of one or more larger sample sized datasets would be an important application of Pantry, and we now mention the promise of such future research in the Discussion (line 426).

Overall the work is thorough and well presented.

We thank the reviewer for their comments.

Reviewer #2 (Remarks on code availability):

I've only reviewed this briefly but it looks well documented and maintained.

Reviewer #3 (Remarks to the Author):

In this well-written manuscript, Munro et al. present a suite of computational tools (Pantry/Pheast) for the generation and genetic analysis of multiple RNA transcriptional modalities. The authors apply their tools to generate multimodal data from Geuvadis and GTEx. Subsequent QTL mapping of the resulting data in these two studies identified a large collection of genes possessing variants with significant effect on modalities distinct from total gene expression. The authors also used the resulting multimodal data to perform TWAS on over 100 complex human traits and showed the use of multimodal data greatly increased gene discovery over the use of total gene expression alone; thereby suggesting that the study of additional modalities can elucidate the origins of human traits and diseases. The paper has many strengths. The Pantry framework likely will have substantial value in a variety of existing studies of gene regulation (with the new multimodal data generated for such studies also having value in related projects like TWAS). I also appreciated the authors making the Geuvadis/GTEx multimodal data available (along with the QTL and TWAS results) to facilitate further analyses based on these popular datasets.

Thank you for the summary and for noting the strengths of the work and data availability.

Main Comments:

The methodology underlying the xQTL mapping procedure requires quite a bit more detail as it is difficult to evaluate. Specifically, it is unclear how such mapping is performed when considering multiple phenotypes (either the cross-modality procedure or the analysis of a specific modality with multiple phenotypes per gene) and how the correlation among related phenotypes are considered within analysis. The manuscript mentions their mapping method is based on tensorQTL, which I believe is a GPU implementation of FastQTL that is designed for a single phenotype (rather than multiple). A supplemental section providing the technical details of model fitting would be helpful.

We have added a more explicit description to the Results (line 105) of tensorQTL's handling of multiple phenotypes per gene, which is enabled with the "--phenotype_groups" argument. We have also added a new section to the Supplementary Information ("Cross-modality mapping with tensorQTL") with technical details on how we used tensorQTL's stepwise regression, phenotype groups, and permutation testing features for cross-modality xQTL mapping. We included a link to the tensorQTL repository (<https://github.com/broadinstitute/tensorqtl>) containing the code and documentation of these features.

Minor Comments:

Lines 147-148: For the concordance analysis, what proportion of variant-RNA phenotype pairs were eligible to be tested for each phenotype in GTEx LCL data?

This was a good suggestion. We have added to the text the number (21,345) and fraction (78%) of significant xQTLs in Geuvadis for which the same variant-phenotype pair was tested in GTEx LCL xQTL mapping and thus used for this concordance analysis.

Lines 219-221: For TWAS analysis assuming a specific modality with multiple phenotypes, was the model trained separately for each phenotype? Would a training model that leveraged multiple phenotypes from the modality (in the spirit of Feng et al. PLoS Genet 17(4): e1008973 or other related methods) perhaps be more powerful?

That is correct, a separate model was trained for each phenotype, so for modalities with multiple phenotypes per gene, there were multiple models per gene. We have added a more explicit description of this on line 229 (“For modalities with multiple RNA phenotypes per gene, this produced multiple models per gene”). As the reviewer suggests, a method that jointly models multiple phenotypes has the potential for more powerful association testing of Pantry phenotypes. However, such a method would need to handle the complexity of multimodal RNA phenotypes, and we believe this would need to be developed in a separate study. We have added discussion of this on line 399:

“While Pantry currently tests RNA phenotypes individually for TWAS associations, a method that jointly models multiple phenotypes has the potential for more powerful association testing. However, such a method would need to handle the complexity of multimodal RNA phenotypes, either by working with potential differences in measurement error across the modalities, or by testing each modality separately such that the results can be interpreted in a comparable way across modalities. Further research is needed to develop joint multimodal xTWAS methods for integration into Pantry.”

Lines 225-227: Of the 4,487 trait-gene pairs that were significant using TWAS trained on Geuvadis, what percentage were also significant using TWAS trained on GTEx LCL data?

This is a good suggestion to measure concordance of TWAS results. There were 2,179 trait-gene pairs significant in both Geuvadis and GTEx LCL, which is 49% of the 4,487 Geuvadis (sample size = 445) significant pairs and 63% of the 3,450 GTEx LCL (sample size = 147) significant pairs. We have added these statistics on line 267.

Figure 4C: Was any modality depleted/enriched for colocalization of TWAS associations in any of the GTEx tissues?

We examined the fraction of TWAS associations per modality per tissue that have strong colocalization evidence, similar to the Figure 4B inset but for all GTEx tissues rather than Geuvadis. The most noticeable factor associated with colocalization fraction was that tissues with very low sample size, in particular kidney cortex (N=73), had especially low fractions for all

modalities. We hypothesize that this is a result of low power to detect both TWAS associations and evidence of variant-level colocalization. Beyond the several tissues with the lowest sample size, though, there is little or no continuation of this trend, and the colocalization fractions converge at around one-third. We have added a two-panel supplementary figure (Supp Figure S6) to show these colocalization fractions in relation to sample size and number of TWAS associations:

Lines 396-399: Why not adjust for gender in the xQTL and xTWAS analyses?

This is a good question, and we have now added an analysis and supplementary figure to address it. Since genetic studies employ a variety of strategies for choosing a set of metadata- and/or data-derived covariates, we tested the effect of adding sex as a covariate along with the genotype and phenotype principal components (PCs). We ran cis-xQTL mapping for each of the six modalities for one GTEx tissue, subcutaneous adipose, without (No-SC) and with (SC) the additional sex covariate. The percent change in number of discovered xQTLs ranged from -0.76% to 0.82% per modality, and 98.3-99.1% of the discovered xGenes with No-SC were also discovered with SC. Among the 87.8-88.7% of the No-SC xQTLs with exact phenotype-top-variant pair matches with SC xQTLs, the Z-scores had Pearson correlations of 0.991 to 0.9997 per modality. Since the addition of a sex covariate had little impact on the results, we decided to use only data-derived PCs for covariates to reduce the complexity of preparing new datasets for Pantry. We made this the default for Pantry and therefore used it for the analyses in this study. That said, Pantry's covariates module can be customized to add covariates from metadata if desired. We have added this analysis and explanation of Pantry's covariates to the Methods (line 461) and refer to a new supplementary figure S9 (below) showing the Z-score correlations. Since the age of each individual is another common metadata-derived covariate, we conducted the same analysis for age and included it in the figure, finding a similarly small impact of adding age as a covariate.

Adding sex to PC covariates

Adding age to PC covariates

REVIEWERS' COMMENTS

Reviewer #1 (Remarks to the Author):

My concerns are satisfactorily addressed.

Reviewer #2 (Remarks to the Author):

The authors have thoroughly addressed the reviewer concerns. I recommend acceptance.

Reviewer #3 (Remarks to the Author):

I have no additional comments. Congratulations to the authors on an impressive work.